# GenoType CM Direct^®^ and VisionArray Myco^®^ for the Rapid Identification of Mycobacteria from Clinical Specimens

**DOI:** 10.3390/jcm11092404

**Published:** 2022-04-25

**Authors:** Hans-Ulrich Schildhaus, Mathis Steindor, Bernd Kölsch, Thomas Herold, Jan Buer, Jan Kehrmann

**Affiliations:** 1Institute of Pathology, University Hospital Essen, University of Duisburg-Essen, 45147 Essen, Germany; hans-ulrich.schildhaus@uk-essen.de (H.-U.S.); bernd.koelsch@uk-essen.de (B.K.); thomas.herold@uk-essen.de (T.H.); 2Department of Pediatric Pulmonology and Sleep Medicine, Children’s Hospital, University of Duisburg-Essen, 45147 Essen, Germany; mathis.steindor@uk-essen.de; 3Institute of Medical Microbiology, University Hospital Essen, University of Duisburg-Essen, 45147 Essen, Germany; jan.buer@uk-essen.de

**Keywords:** NTM, non-tuberculous mycobacteria, rapid diagnostic test, molecular identification

## Abstract

*M. tuberculosis* is the single infectious agent responsible for most deaths worldwide outside of pandemics. Diseases due to non-tuberculous mycobacteria (NTM) are increasing in many regions of the world. The two molecular assays GenoType CM direct^®^ (GTCMd) (Bruker, Billerica, MA, USA) and VisionArray Myco^®^ (VAM) (ZytoVision, Bremerhaven, Germany) are based on the DNA/DNA hybridization technique, and allow for the identification of tuberculous and the most clinically relevant non-tuberculous mycobacterial species from clinical specimens. We evaluated the performance of both assays for the identification of mycobacteria from 65 clinical specimens of 65 patients and compared it with the results of conventional culture. Based on conventional culture that recovered 37 mycobacterial isolates including 11 tuberculous and 26 NTM isolates, sensitivity, specificity, positive predictive value and negative predictive value were 89.2%, 81.5%, 86.8% and 84.6% for GTCMd and 73.0%, 96.3%, 96.4% and 72.2% for VAM. Additionally, GTCMd identified mycobacteria from five and VAM from one culture-negative sample. Both assays identified a mycobacterium in one sample overgrown by other microorganisms. Two *M. abscessus* subsp. *abscessus* isolates grown from culture were identified as *M. chelonae* by GTCMd assay. In conclusion, both assays improve the rapid identification of mycobacteria directly from clinical specimens.

## 1. Introduction

The genus *Mycobacterium* comprises more than 190 different species and subspecies (www.bacterio.net/mycobacterium.html (accessed 18 March 2022)). According to the WHO, *Mycobacterium tuberculosis* was the pathogen responsible for most deaths worldwide from a single infectious agent before the COVID-19 pandemic [1]. The molecular detection directly from clinical specimens has substantially improved diagnostics toward rapid and sensitive identification of tuberculous mycobacteria as compared to microscopy of acid fast bacilli (AFB) [2].

The incidence of disease caused by non-tuberculous mycobacteria (NTM) has increased globally over the last three decades and NTM is increasingly isolated in many clinical microbiology laboratories [3,4,5,6]. The pathogenicity of NTM varies between species and ranges from those that are rarely caused by human NTM disease, like *M. gordonae*, to species that are frequently associated with NTM disease like *M. kansasii* [7]. The identification of NTM to the species level supports assessing the clinical relevance and directing antimicrobial treatment [8]. While molecular tests for the detection of tuberculous mycobacteria directly from clinical specimens are established globally for many years, most molecular tests available for identification of NTM from clinical specimens do not allow for species identification or are limited to identifying members of the *M. avium* complex (MAC) [9,10,11,12,13]. The two molecular assays GenoType CM direct^®^ (GTCMd) (Bruker, Billerica, MA, USA) and Vision Array Myco^®^ (VAM) (ZytoVision, Bremerhaven, Germany) allow for the identification of tuberculous mycobacteria and a wide spectrum of the most important clinically relevant NTM species directly from clinical specimens using the DNA/DNA hybridization technique. To our knowledge, this is the first study to evaluate the performance of both assays for the identification of tuberculous and non-tuberculous mycobacteria directly from clinical specimens of patients with the suspected mycobacterial disease and compared them with the results obtained by conventional culture.

## 2. Materials and Methods

### 2.1. Study Population

A total of 65 primary specimens of 65 patients of the University Medicine Essen with the clinically suspected mycobacterial disease were included in our study. The Ethics Committee of the Medical Faculty of the University of Duisburg-Essen reviewed and approved our study (No. 22-10525-BO). All samples were taken as a part of standard care procedures. We followed the recommendations of the ICH-GCPC guidelines and the study was performed in accordance with the latest version of the Declaration of Helsinki.

### 2.2. Sample Processing for DNA Extraction, Microscopy and Mycobacterial Culture

Primary clinical specimens were decontaminated with N-acetyl-L-cysteine-sodium hydroxide (NALC-NaOH) using the BBL MycoPrep^TM^ Mycobacterial System Digestion/Decontamination Kit (Becton Dickinson (BD), Franklin Lakes, NJ, USA). After neutralization with phosphate buffer, samples were centrifuged at 3300× *g*. The sediment was resuspended in 1–1.5 mL phosphate buffer. Auramin staining was performed with 10 µL of decontaminated samples using the Aerospray/TB^®^ series 2 slide stainer (Kreienbaum, Langenfeld, Germany) and assessed as follows: more than 10 acid fast bacilli (AFB) per field at 1000× magnification were classified as +++, between 1 to 10 AFB per field after examination of 100 fields were classified as ++, between 10 to 99 AFB per 100 fields were classified as +, between 1 AFB per 300 fields to 9 AFB per 100 fields were classified as +/− and no AFB per 300 fields were classified as negative. We inoculated 500 µL of decontaminated samples into MGIT liquid broth (BD), 50–100 µL each on Loewenstein Jensen and Stonebrink solid media (Oxoid, Wesel, Germany). Inoculated media were cultivated at 36 °C ± 1 °C at aerobic atmosphere, liquid cultures for a period of six weeks, and solid cultures for eight weeks. Samples of cystic fibrosis (CF) patients and tissue samples were additionally cultivated in MGIT liquid broth and Loewenstein Jensen and Stonebrink solid media at 30 °C. In case of overgrowth of cultures with bacterial or fungal pathogens of cultures from patients with CF, a sequential second decontamination procedure was performed using 5% oxalic acid. Identification of mycobacteria from positive cultures was performed by GenoType CM and GenoType AS (both Bruker) following the manufacturer’s manual. For mycobacteria that were identified as *M. intracellulare* complex or *M. abscessus* complex by GenoType CM, GenoType NTM-DR (Bruker) was performed subsequently to identify the *M. abscessus* subspecies or to discriminate *M. chimaera* from *M. intracellulare*. All cultured isolates that were identified as *M. tuberculosis* complex were identified to the species level using the GenoType MTBC (Bruker).

### 2.3. DNA Extraction and Implementation of GTCMd and VAM Molecular Assays

DNA was extracted from decontaminated patient samples using GenoLyse (Bruker) according to the manufacturer´s recommendations. We used 5 µL of the same DNA solution for amplicon PCR of GTCMd assay and 2.5 to 5 µL for VAM, that were both performed according to the manufacturer´s manual. The GTCMd assay is approved to identify *M. tuberculosis* complex, *M. avium*, *M. intracellulare* complex, *M. absessus* complex, *M. chelonae*, *M. kansasii*, *M. fortuitum* group, *M. gordonae*, *M. scrofulaceum*/*M. intracellulare*, *M. malmoense*, *M. interjectum*, *M. szulgai*, *M. marinum*/*M. ulcerans*, and *M. xenopi*. The VAM assay Version 2.0 (ZytoVision, Bremerhaven, Germany), which was used for 28 samples, identifies *M. tuberculosis* complex, *M. avium* complex, and *M. chimaera* separately from *M. avium* complex, *M. absessus* complex, *M. chelonae*, *M. kansasii*, *M. fortuitum*, *M. gordonae*, *M. scrofulaceum*/*M. parascrofulaceum*, *M. malmoense*, *M. haemophilum*, *M. genavense*, *M. marinum*/*M. ulcerans*, *M. smegmatis*, *M. simiae*, and *M. xenopi*. The VAM version 1.0, which was used for the first 37 samples included in the study differs from the 2.0 Version as it did not identify *M. chimaera* and *M. gordonae* but reported M. spec. for genus-level identification. Both assays do not differentiate members within the *M. tuberculosis* complex and *M. abscessus* complex.

The amplification of human DNA from sample specimens indicates that the PCR is not inhibited. As this study included samples with low amounts of human DNA, e.g., from bronchoalveolar fluid specimens, and no human DNA was added during preparation, no PCR products were present for some samples. When no PCR products were present, the sample was evaluated as negative for the presence of mycobacteria for the VAM assay.

### 2.4. Statistics

Sensitivity was calculated by the number of positive test results that were in agreement with culture results/total number of positive culture results × 100. Specificity was calculated by the number of negative test results that were in agreement with negative culture results/total number of negative culture results × 100. PPV was calculated by the number of positive test results that were in agreement with culture results/total number of positive test results × 100. NPV was calculated by the number of negative test results that were in agreement with culture results/total number of negative test results × 100.

Fisher´s exact test was used to calculate associations between test results and culture results for both assays. A *p*-value < 0.05 was considered as statistically significant.

## 3. Results

Between May 2019 and November 2021, we included 65 patients with the suspected mycobacterial disease, comprising 13 patients with CF. Of 65 specimens, 54 were respiratory specimens, including 25 sputum samples, 16 bronchoalveolar lavage fluid, and 13 bronchial aspirate specimens (Table 1). In addition, 8 tissue samples (3 lymph node specimens, 3 lung specimens, one gut, and one skin tissue sample), two swab samples (one brain and one bone swab), and one heparin blood specimen taken from a central venous port device were included. Microscopy for detection of AFB from primary specimens was positive for 38 specimens and classified as +/− for 9, + for 20, ++ for 6, and +++ for 3 specimens and was negative for 27 specimens.

We recovered 37 mycobacterial isolates from the cultures of 37 patients, while mycobacterial culture was negative for 27 patients. In addition, one sputum culture was overgrown by bacteria other than mycobacteria, limiting the significance of this culture. The sensitivity of the performance of molecular tests is dependent on the number of mycobacteria present in the primary samples, which was semi-quantitatively assessed by microscopy of AFB. Both molecular assays identified three of three specimens (100%) correctly to the species level that were classified as +++ by microscopy for AFB and 5 of 6 (83%) that were classified as ++ (Table 2). The performance for primary specimens classified as + was 95% (19/20) for GTCMd and 85% for VAM (17/20), for specimens classified as +/− 60% (3/5) for GTCMd and 40% (2 of 5) for VAM, and for specimens classified as negative for AFB was 100% (3/3) for GTCMd and 0% for VAM.

Tuberculous mycobacteria were cultured from 11 specimens, including one *M. bovis* and 10 *M. tuberculosis* isolates (Table 3). *M. tuberculosis* was the species most frequently cultured, followed by *M. avium* (6), *M. chimaera* (5), *M. abscessus* subsp. *abscessus* (5), *M. intracellulare* (3), *M. kansasii* (3), *M. abscessus* subsp. *massiliense* (2), and *M. xenopi* (2) (Table 3). The GTCMd assay correctly identified 33 of 37 cultured mycobacterial isolates to the species level, resulting in a sensitivity of 89% compared with culture. The sensitivity of the VAM was 73%, and 27 of 37 mycobacteria recovered from conventional culture were identified correctly by the assay from primary specimens. Of the ten patient samples not identified correctly, three isolates, one positive with *M. kansasii*, one with *M. abscessus* subsp. *abscessus* and one with *M. chimaera*, were identified correctly at the genus level. The assay did not detect mycobacteria for 7 specimens that were culture-positive, two for *M. abscessus* subsp. *abscessus*, two for *M. tuberculosis*, one for *M. intracellulare,* one for *M. chimaera,* and one for *M. kansasii*. Auramin stain of these isolates was classified as negative for three, +/− for two, + for one, and ++ for one isolate (Table 4).

Of the four mycobacterial species recovered from culture and not identified by the GTCMd assay to the species level, one *M. tuberculosis* complex isolate and one *M. intracellulare* isolate were identified to the genus level, both classified as +/− by microscopy. Two *M. abscessus* subsp. *abscessus* isolates were identified as *M. chelonae* (Table 4), a species closely related to *M. abscessus*.

The GTCMd assay additionally identified mycobacteria from five culture-negative specimens, *M. kansasii* and *M. xenopi* from two specimens each. Additionally, no single mycobacterial species could be assigned for one culture-negative specimen, showing the combination of bands 4, 5, and 10 for which no individual species is reported by the manufacturer, but might indicate the simultaneous presence of *M. avium* (band 4) and *M. chelonae* (combination of band 5 and 10) in this specimen. GTCMd also detected *M. chelonae* from the specimen that was overgrown by bacteria other than mycobacteria and was classified as +/− in the Auramin stain, while VAM was positive for *M. smegmatis* from this specimen. A borderline positive result for *M. simiae* was obtained by VAM from one culture-negative specimen that was classified as +/− by microscopy for AFB of the primary specimen. *M. simiae* cannot be identified by the GTCMd assay.

Compared with the results of conventional culture, the GTCMd assayreported negative results for 22 of 27 specimens that were culture negative and the VAM yielded negative results for 26 of 27 culture-negative specimens. The test performance for sensitivity, specificity, positive predictive value and negative predictive value compared with culture results were 89.2%, 81.5% 86.8%, and 84.6% for GTCMd^®^ and 73.0%, 96.3%, 96.4% and 72.2% for VAM (Table 5). Fisher’s exact test showed an association between the results of the GTCMd and the culture results (*p* < 0.0001), and between the results of VAM and the culture results (*p* < 0.0001).

## 4. Discussion

To our knowledge, this is the first study evaluating the performance of the GTCMd and VAM for identification of mycobacteria directly from clinical specimens. Overall, we found that both assays were valuable for the rapid identification of mycobacteria from clinical specimens with a higher sensitivity of 89.2% for the GTCMd and a higher specificity of 96.3% for the VAM assay for identification of mycobacteria to the species level related to the results of the mycobacterial culture. Both assays exhibited a good sensitivity for identifying cultured mycobacterial isolates from primary specimens classified as ++ and +++ for AFB. However, the performance for identification of mycobacteria from specimens classified as +, +/−, and negative for detection for AFB was higher for the GTCMd assay^®^, indicating a lower limit of detection for this assay as compared to the VAM. The limit of detection for the GenoType CM^®^ as reported by the instructions for use of the manufacturer varies between 750 CFU/mL for *M. fortuitum* and 50,000 CFU/mL for *M. intracellulare*. Interestingly, one of the four isolates not identified by GTCMd was one *M. intracellulare* from a specimen classified +/− in microscopy, supporting the limited sensitivity for this species as reported by the manufacturer. Two of the four isolates not identified were *M. abscessus* subsp. *abscessus* isolates that were identified as *M. chelonae*, a species closely related to *M. abscessus* [14]. Both were classified as separate species since 1992 and differ in response to antimicrobial treatment [15]. The banding pattern of the GTCMd assay discriminates both species only by the presence of the band no 6, which is visible only in *M. abscessus* (presence of bands 5, 6, and 10 instead of 5 and 10 for M. *chelonae*)*,* and may be only weakly established as stated by the manufacturer. In our study, band 6 was not visible for these two *M. abscessus* isolates, suggesting that the performance of the assay for discriminating both species from clinical samples may be limited for some isolates. As one of these isolates was classified as ++ for microscopy of AFB, the low number of bacteria in the sample is unlikely to explain the negative presence of this band, as the limit of detection is reported to be 1500 CFU/mL for *M. abscessus* by the manufacturer.

The sensitivity for the detection of tuberculous mycobacteria was 91% (10 of 11) for the GTCMd and 82% (9 of 11) for the VAM. Both tests did not report the presence of MTB complex from one specimen each that exhibited AFB by microscopy. As primary specimens of most MTB complex isolates were positive for AFB, the performance of the tests for the detection of tuberculous mycobacteria needs further evaluation and should be compared with more MTB complex samples exhibiting negative microscopy for AFB to compare them with the existing sensitive assays for detection of tuberculous mycobacteria.

While the rapid identification of mycobacteria of the *M. tuberculosis* complex using molecular assays has significantly accelerated and improved the diagnostic of tuberculosis worldwide, the use of molecular assays for the identification of NTM directly from clinical specimens without subsequent sequencing is limited to date and does not allow for identification of different NTM species [9].

In contrast to tuberculous mycobacteria, the significance of molecular assays for supporting the diagnosis of NTM disease is less obvious. The isolation of NTM from clinical specimens alone is not sufficient for the diagnosis of NTM disease. The significance of NTM recovered from clinical specimens from individuals that meet the clinical and radiological criteria for NTM disease is interpreted in the context of the number of positive cultures and the specific species isolated [8]. The GTCMd and VAM both include the most important NTM to cause human disease, including *M. avium* complex, *M. kansasii*, *M. abscessus* complex, and *M. xenopi*. Molecular methods used for the detection of mycobacteria from clinical specimens can not only accelerate mycobacterial identification but may increase the sensitivity of NTM detected by conventional culture because of several reasons: First, molecular assays are independent of active bacterial growth and may be advantageous, especially for fastidious organisms that rely on the presence of additional growth factors like *M. haemophilum* or *M. genavense,* that both can be identified using VAM.

Second, mycobacterial growth may also be impeded when cultures are not incubated at the optimal growth temperature, which is 35–37 °C for many clinically relevant mycobacteria but may be 30 °C or less for some clinically relevant mycobacteria such as *M. marinum*, *M. ulcerans* and *M. haemophilum* [7] or 42 °C to 45 °C for the thermophile species *M. xenopi* [16], temperatures not routinely used for the incubation of all mycobacterial cultures.

Third, decontamination procedures, routinely performed for mycobacterial cultures, may also impair mycobacterial growth. Especially CF patients are often colonized with multidrug-resistant bacteria such as *P. aeruginosa*, which may contaminate mycobacterial cultures. An additional sequential decontamination performed in patients with CF using 5% oxalic acid inhibits the growth of *P. aeruginosa* and other commensal bacteria but may negatively also affect mycobacterial growth [17]. GTCMd identified *M. xenopi* and *M. kansasii* each from two separate culture-negative specimens. *M. xenopi* is one of the slow-growing mycobacteria that benefits from an increased incubation period, with an increased number of isolates have been detected after 42 days of incubation of MGIT liquid cultures [16]. Nevertheless, it is difficult to conclude, whether decontamination procedures and/or cultivation at suboptimal growth temperature for *M. xenopi* negatively affected mycobacterial growth in these culture-negative specimens. Although *M. xenopi* may be a causative pathogen for NTM lung disease, also pseudoinfections of contaminated specimens collected from bronchoscopes, have been reported [18]. In our study, the four culture-negative specimens that were positive for *M. xenopi* and *M. kansasii* and the one sample exhibiting positive bands 4, 5, and 10 in GTCMd were all taken during bronchoscopy. In addition, one borderline result for *M. simiae* from a culture-negative specimen was identified by VAM and *M. smegmatis* was detected from the overgrown specimen that was classified as +/− in microscopy. An assessment, especially for the overgrown specimen, is limited, as *M. smegmatis* but also *M. simiae* are not identified by the GTCMd assay that identified *M. chelonae* from this specimen.

Our study was limited by its single-center design from a European Tertiary care setting. The patient spectrum may not be representative of regions from other continents. Patients with CF were overrepresented making up 20% of the total patients included in our study. CF patients have a much higher risk for NTM infections than the general population and NTM prevalence is reported to range between 13 and 23% [19]. In addition, our study was limited by the number of specimens included and does not allow for assessing the performance of the tests for all species that can be identified by the molecular assays. However, in our study, both assays have shown to be valuable to accelerate and improve rapid identification of NTM from clinical specimens of patients from a European tertiary care setting, identifying the majority of mycobacteria from specimens that were positive for AFB in microscopy.

## 5. Conclusions

The two molecular assays GTCMd and VAM substantially extend the number of clinically relevant NTM species that can be rapidly identified from clinical specimens by molecular tests. Both tests may be valuable tools in clinical microbiology laboratories for accelerating and improving the identification of NTM from clinical specimens.

## Figures and Tables

**Table 1 jcm-11-02404-t001:** Number of sample specimens and quantification of acid fast bacilli.

	No.	Microscopy				
Specimen	Total	Negative	+/−	+	++	+++
Sputum	25	6	5	11	3	0
BALF	16	13	3	0	0	0
Bronchial fluid	13	3	0	7	2	1
Tissue	8	3	1	2	1	1
Swab	2	1	0	0	0	1
Blood	1	1	0	0	0	0

**Table 2 jcm-11-02404-t002:** Correct results of molecular assays for identification of mycobacterial species isolated from culture referring to the classification of the quantity of acid fast bacilli assessed by microscopy (*n* = 37).

Microscopy	Negative	+/−	+	++	+++
GTCMd	3/3 (100)	3/5 (60)	19/20 (95)	5/6 (83)	3/3 (100)
VAM	0/3 (0)	2/5 (40)	17/20 (85)	5/6 (83)	3/3 (100)

**Table 3 jcm-11-02404-t003:** Performance for correct identification of different mycobacterial species cultured (*n* = 37).

		Sensitivity	
Culture	Number	GTCMd	VAM
*M. chimaera*	7	7/7 (100)	5/7 (71)
*M. avium*	6	6/6 (100)	6/6 (100)
*Mabsc.* subsp. *abscessus*	5	3/5 (60)	3/5 (60)
*M. kansasii*	3	3/3 (100)	1/3 (33)
*M. xenopi*	2	2/2 (100)	2/2 (100)
*Mabsc.* subsp. *massiliense*	2	2/2 (100)	1/2 (50)
*M. intracellulare*	1	0/1 (0)	0/1 (0)
*M. tuberculosis*	10	9/10 (90)	8/10 (80)
*M. bovis*	1	1/1 (100)	1/1 (100)
**Total**	**37**	**33/37 (89.2)**	**27/37 (73)**

**Table 4 jcm-11-02404-t004:** Discrepant results between conventional culture and molecular tests.

Microscopy	Culture Result	GTCMd	VAM
+/−	*M. chimaera*	*M. intracell. complex*	**Negative**
−	*M. chimaera*	*M. intracell. complex*	**M. spec.**
+/−	*M. intracellulare*	**M. spec.**	**no PCR product**
+	*M. kansasii*	*M. kansasii*	**M. spec.**
−	*M. kansasii*	*M. kansasii*	**no PCR product**
−	*M. tuberculosis*	*MTBC*	**no PCR product**
+/−	*M. tuberculosis*	**M. spec.**	MTBC
+	*M. tuberculosis*	*MTBC*	**Negative**
+	*M. absc.* subsp. *absc.*	*M. absc.* complex	** *M. spec/M. genavense* **
++	*M. absc.* subsp. *absc.*	** *M. chelonae* **	**Negative**
+	*M. absc.* subsp. *absc.*	** *M. chelonae* **	*M. absc.* complex
+/−	***M. absc.* subsp. *mass.***	*M. absc.* complex	**Negative**
−	Negative	** *M. kansasii* **	no PCR product
−	Negative	** *M. kansasii* **	no PCR product
−	Negative	** *M. xenopi* **	no PCR product
−	Negative	** *M. xenopi* **	no PCR product
−	Negative	**band 4,5,10 (weak)**	no PCR product
+/−	Negative	Negative	** *M. simiae* **
+/−	Contaminated	** *M. chelonae* **	*M. smegmatis*

Bold font indicates discrepant results.

**Table 5 jcm-11-02404-t005:** Two-by-two tables illustrating the results of GTCMd assay and VAM assay compared to culture results to assess sensitivity, specificity, PPV and NPV.

		Culture	
		Positive	Negative
**GTCMd**	Positive	33	5 ^b^
	Negative	4 ^a^	22
		**Culture**	
		**Positive**	**Negative**
**VAM**	Positive	27	1 ^d^
	Negative	10 ^c^	26

^a^ 1 × *M. intracellulare* and 1 × *M. tuberculosis* complex grown from culture were identified as M. spec. by GTCMd, 2 × *M. abscessus* subsp. *abscessus* were identified as *M. chelonae* by GTCMd. ^b^ 5 culture-negative samples were identified as 2 × *M. xenopi*, 2 × *M. kansasii* and 1 × banding pattern 4, 5, 10 with no identification of a single NTM species by GTCMd. ^c^ 1 × *M. intracellulare*, 1 × *M. abscessus* subsp. *abscessus*, 1 × *M. kansasii* grown from specimens were identified as M. spec., 7 isolates from seven culture-positive specimens were unidentified (2 × *M. tuberculosis*, 2 × *M. chimaera*, 1 × *M. abscessus* subsp. *abscessus*, 1 × *M. abscessus* subsp. *massiliense*, 1 × *M. kansasii*). ^d^ 1 × *M. simiae* (borderline) from 1 culture-negative specimen.

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
