# Peer review of "GenoType CM Direct® and VisionArray Myco® for the Rapid Identification of Mycobacteria from Clinical Specimens"

_jcm, 2022, doi:10.3390/jcm11092404_

Round 1

Reviewer 1 Report

Clear study comparing two molecular assays for detection of tuberculous and non-tuberculous mycobacteria from clinical samples. Although the sample size is modest for a the purpose of evaluating a diagnostic, the lack of studies for this evaluation makes this contribution worth consideration. 

Minor typos to be corrected, as indicated in pdf.

Author Response

Comments and Suggestions for Authors:

Clear study comparing two molecular assays for detection of tuberculous and non-tuberculous mycobacteria from clinical samples. Although the sample size is modest for a the purpose of evaluating a diagnostic, the lack of studies for this evaluation makes this contribution worth consideration. 

Minor typos to be corrected, as indicated in pdf.

Answer from the authors: We thank the reviewer for the positive and careful review. We have now corrected all typing errors indicated in the pdf.

Reviewer 2 Report

The present manuscript provides a comparison among the performances of two molecular diagnostic tests, focusing on the test ability to identify non-tuberculous mycobacteria. Overall, limitations of the study are properly identified and the article is well-written.

In reporting performances, only descriptive statistics are used. Some additional statistics may be required.

Author Response

Comments and Suggestions for Authors:

The present manuscript provides a comparison among the performances of two molecular diagnostic tests, focusing on the test ability to identify non-tuberculous mycobacteria. Overall, limitations of the study are properly identified and the article is well-written.

In reporting performances, only descriptive statistics are used. Some additional statistics may be required.

Answer from the authors: We appreciate the positive review of the reviewer. We have now additionally calculated whether the associations between the results of the assays and the culture results are significant using the Fisher´s exact test. The results of both assays, the GTCMd and VAM are significantly associated with the culture results with p<0.0001 for both assays (lines 186-188). We have now also added a paragraph to the Materials and Methods section describing the statistics performed in the study (lines 111-120).